# Ultrahigh thermoelectric power factor in flexible hybrid inorganic-organic superlattice

Chunlei Wan[1], Ruoming Tian[2], Mami Kondou[3], Ronggui Yang[4,5,6], Pengan Zong[1] & Kunihito Koumoto[2]

Hybrid inorganic–organic superlattice with an electron-transmitting but phonon-blocking structure has emerged as a promising flexible thin film thermoelectric material. However, the substantial challenge in optimizing carrier concentration without disrupting the superlattice structure prevents further improvement of the thermoelectric performance. Here we demonstrate a strategy for carrier optimization in a hybrid inorganic–organic superlattice of $TiS_2$[tetrabutylammonium]$_x$[hexylammonium]$_y$, where the organic layers are composed of a random mixture of tetrabutylammonium and hexylammonium molecules. By vacuum heating the hybrid materials at an intermediate temperature, the hexylammonium molecules with a lower boiling point are selectively de-intercalated, which reduces the electron density due to the requirement of electroneutrality. The tetrabutylammonium molecules with a higher boiling point remain to support and stabilize the superlattice structure. The carrier concentration can thus be effectively reduced, resulting in a remarkably high power factor of $904\,\mu W\,m^{-1}\,K^{-2}$ at 300 K for flexible thermoelectrics, approaching the values achieved in conventional inorganic semiconductors.

[1] State Key Laboratory of New Ceramics and Fine Processing, School of Materials Science and Engineering, Tsinghua University, Beijing 100084, China. [2] Toyota Physical and Chemical Research Institute, Nagakute 480-1192, Japan. [3] Graduate School of Engineering, Nagoya University, Nagoya 464-8603, Japan. [4] Department of Mechanical Engineering, University of Colorado, Boulder, CO 80309, USA. [5] Materials Science and Engineering Program, University of Colorado, Boulder, CO 80309, USA. [6] Buildings and Thermal Systems Center, National Renewable Energy Laboratory, Golden, CO 80401, USA. Correspondence and requests for materials should be addressed to C.W. (email: wancl@mail.tsinghua.edu.cn)

There is a rapidly growing interest in flexible thin film thermoelectrics[1] for ambient temperature cooling and power generation on the microwatt-to-watt scale, where temperature gradients are moderate, surfaces are irregular, conventional thermoelectric materials with toxic or rare elements are prohibited, and low cost is preferred[2–6]. Especially, these flexible thin film thermoelectric devices are now under active pursuit for wearable energy harvesting, as they can generate electricity to power sensors by the temperature differences between human body and the environmental atmosphere[3]. These thermoelectric generators have many merits over the widely used lithium ion batteries, such as safety, long-lasting and maintenance-free, which is considered as a promising wearable power source[3].

Conventional high-efficiency thermoelectric materials, such as $Bi_2Te_3$ and Skutterudites, are usually covalent- or ionic-bonded, and rigid, which is difficult to make devices mechanically flexible[7]. In contrast, conducting polymers and polymer composites are much more advantageous for making flexible thermoelectric devices, which combines structural flexibility and solution processability[1]. The thermoelectric power factor reached $469\,\mu W\,m^{-1}\,K^{-2}$ for pure PEDOT:PSS (poly(3,4-ethylenedioxythiophene): polystyrene sulfonate)[8] and recently elevated to $1825\,\mu W\,m^{-1}\,K^{-2}$ for PANI/graphene/PANI/DWNT nanocomposites (PANI: polyaniline, DWNT: double-walled carbon nanotubes)[9]. However, these organic thermoelectric materials are mainly p-type ones while their n-type counterpart is still lacking but in need for the completion of a high-efficiency flexible thermoelectric module. It is very difficult to make high performance n-type organic thermoelectric materials due to the low electron affinity of organic materials. For example, a low power factor of $0.6\,\mu W\,m^{-1}\,K^{-2}$ was reported for the n-type polymer poly[N,N′-bis(2-octyl-dodecyl)–1,4,5,8-napthalenedicarboximide-2, 6-diyl]-alt-5,5′-(2,2′-bithiophene)] (P(NDIOD-T2)[10] and $1.4\,\mu W\,m^{-1}\,K^{-2}$ was reported for the self-doped perylene diimides (PDI)[11].

To overcome this challenge, inorganic components have been incorporated into polymers to make n-type flexible materials with enhanced thermoelectric performance[12–16]. For example, by embedding metallic $Cu_xBi_2Se_3$ nanoplates into the polyvinylidene fluoride matrix, an n-type power factor of $103\,\mu W\,mK^{-2}$ was obtained[17]. Interfacial doping of bismuth into organic materials shows n-type thermoelectric power factor of $108\,\mu W\,m^{-1}\,K^{-2}$ [18]. Transition metal elements were added into polymers to form organometallic poly[metal-ethylenetetrathiolate] complex, in which through-bond coupling between the metal d-orbitals and ligand π-orbitals induces one-dimensional n-type conductivity. The power factor reached $66\,\mu W\,m^{-1}\,K^{-2}$ in an earlier report[19] and was then improved to $453\,\mu W\,m^{-1}\,K^{-2}$ [20]. However, the power factors of all these n-type polymer composites are still low compared with state-of-the-art inorganic thermoelectric semiconductors, such as $Bi_2Te_3$, PbTe, and Skutterudites.

To further improve the power factor of the n-type flexible thermoelectric materials, there is an emerging direction to make a hybrid inorganic/organic superlattice by stacking metallic/semi-conducting inorganic materials and organic molecules layer by layer. The thermoelectric power factor could reach that of the inorganic materials while the use of the organic molecules results in low thermal conductivity and greater mechanical flexibility. Atomic layer deposition was used to make hybrid $(Zn_{1-x}Al_xO)$/Hydroquinone superlattice with a power factor of $38\,\mu W\,m^{-1}\,K^{-2}$ [21,22]. We have recently demonstrated a hybrid inorganic–organic superlattice of $TiS_2$ monolayers and organic molecules, with a record-high n-type power factor of $450\,\mu W\,m^{-1}\,K^{-2}$ [23,24]. However, the power factor remains to be improved as the carrier concentration is far from being optimized. Conventionally, chemical doping is used in inorganic

materials for carrier concentration optimization by co-firing dopant elements with the pristine material at high temperature. However, this method cannot be applied in the inorganic–organic superlattice, which would decompose before the doping temperature is achieved.

In this paper, we develop a strategy to effectively tune the carrier concentration without disrupting the hybrid superlattice structure. A giant high power factor ($904\,\mu W\,m^{-1}\,K^{-2}$) is obtained for the flexible n-type material at room temperature, which is even close to that of some high-ZT inorganic materials, such as SnSe ($1010\,\mu W\,m^{-1}\,K^{-2}$ @ 850 K)[25] and $Cu_{2-x}Se$ ($1200$ $\mu W\,m^{-1}\,K^{-2}$@ 1000 K)[26].

## Results

**The dependence of the power factor on carrier concentration.** The previously reported hybrid inorganic–organic $TiS_2(HA)_{0.08}$ $(H_2O)_{0.22}(DMSO)_{0.03}$ superlattice by us has a power factor of $450\,\mu W\,m^{-1}\,K^{-2}$ with a very high carrier concentration of $7.6 \times 10^{20}\,cm^{-3}$ [24]. The power factor can be remarkably improved if the carrier concentration was reduced, as shown in Supplementary Fig. 1 and Supplementary Note 1. As mentioned earlier, completely different strategies than chemical dopant co-firing should be developed for tuning carrier concentration in an inorganic–organic material. The hybrid superlattice is synthesized through the electrochemical intercalation process[24]. During this process, electrons are injected into the $TiS_2$ anode, where the negative charges compensate the intercalated organic cations. (Supplementary Fig. 2). Apparently, the electrical neutrality condition between the electron charges in the inorganic material and the organic cations can then be used as the guiding principle to tune the carrier concentration in the hybrid superlattice. In other words, by reducing the organic cation density, the corresponding electron charge inside the inorganic $TiS_2$ layers can be reduced.

**Carrier optimization with mono-molecule intercalation.** The first attempt was made by simply heating inorganic/organic superlattice in vacuum at higher temperatures, where part of the organic molecules is evaporated, which leads to a reduction of the carrier concentration (Fig. 1a). The percentage of the evaporated molecules can be controlled by the heating conditions, such as the temperature and the heating duration. As shown in Supplementary Fig. 3a, the samples were heated for 1 h at temperatures ranging from 100 to 200 °C. The Seebeck coefficient increases with increasing temperature, suggesting a decrease in carrier concentration. According to the estimation (Supplementary Fig. 1 and the Supplementary Note 1), 180 °C was chosen as heating temperature for the rest of the work as the highest Seebeck coefficient ($150\,\mu V\,K^{-1}$) was obtained at this temperature. Then the sample was heated at 180 °C for different durations. It shows that with the increasing time, the Seebeck coefficient quickly increases after 45 min and then saturates even with a duration as long as 3 h. (Supplementary Fig. 3b) Therefore, the heating duration was chosen to be 1 h.

The XRD results in Fig. 1b show a layered structure for both the pristine $TiS_2$ single crystal and the hybrid materials. $TiS_2(HA)_x(DMSO)_y$ is the material after electrochemical intercalation, and $TiS_2(HA)_z$ represent the composition after vacuum heating the $TiS_2(HA)_x(DMSO)_y$ sample. The interlayer distance (the lattice parameter along the c axis) increases from 5.69 Å for the pristine $TiS_2$ to 18.3 Å for the $TiS_2(HA)_x(DMSO)_y$, as both HA and DMSO molecules were intercalated in to the van der Waals gap forming a paraffin-like bilayer structure[27]. After vacuum heating at 180 °C for 1 h, large portion of the organic molecules, including HA and DMSO, were evaporated, which

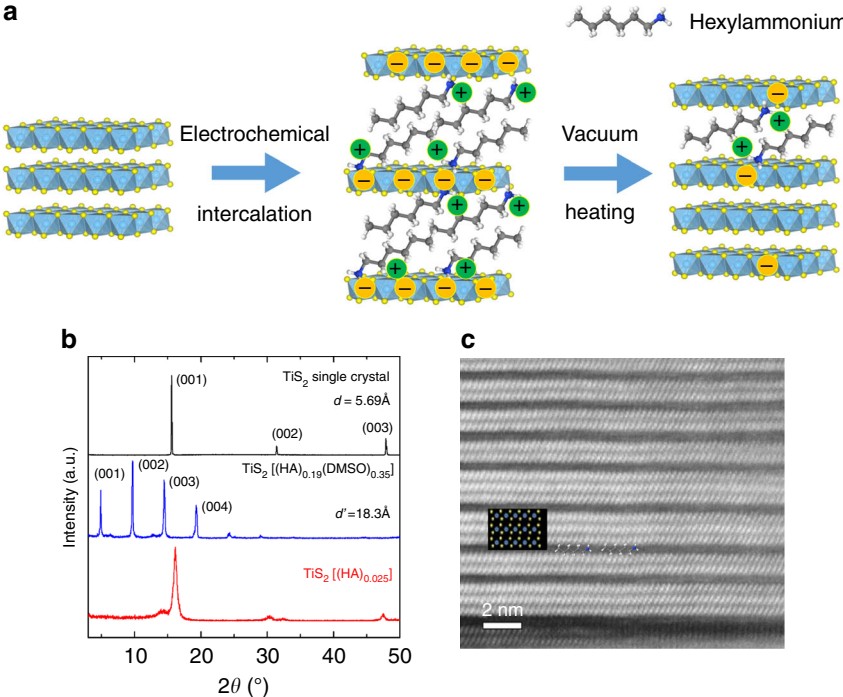

**Fig. 1 a** Evolution of structure and carrier concentration of $TiS_2$ single crystal with the electrochemical intercalation of HA molecules and vacuum heating. **b** XRD patterns of $TiS_2$ single crystal, $TiS_2(HA)_{0.19}(DMSO)_{0.35}$ and $TiS_2(HA)_{0.25}$, **c** Magnified HAADF-STEM image of $TiS_2(HA)_{0.025}$

greatly shrinks the interlayer distance. The XRD pattern almost changes back to that of the original $TiS_2$ single crystal, but the (001) peak were shifted and broadened, suggesting the presence of stage disorder in the hybrid materials[28]. The structure was further confirmed by TEM as shown in Fig. 1c. It shows a disordered high-stage layered structure, in which two or three layers of $TiS_2$ are separated by one organic layer. The results show that the molecules were not uniformly evaporated from the hybrid material. It was shown that in the intercalation compounds, the molecules intercalated into a layered compound increases the strain energy by expanding the interlayer distance[29]. When there are too few molecules to fill in all the possible interlayer space, these molecules tend to aggregate to form clusters between a single set of the layers, rather than randomly and uniformly distribute in all the interlayer space, because this aggregation can reduce the strain energy of the whole system[29]. Therefore, when most of the organic molecules are removed from the $TiS_2(HA)_x(DMSO)_y$ hybrid material by vacuum heating, a random high-stage compound is formed to minimize the strain energy, as seen in Fig. 1c.

Nuclear magnetic resonance (NMR) analysis was performed and the composition for the hybrid material after vacuum heating was determined to be $TiS_2(HA)_{0.025}$, which is consistent with the random staging structure, suggested by the XRD and TEM results. There have been some reports showing that both neutral amine and protonated amine coexist in the interlayer space in an intercalated material[27]. However, in the current approach, organic cation (hexylammonium) dissolved in an aprotic solvent is used as an electrolyte, in which hexylammonium cannot change to neutral amines. Therefore, during the electrochemical intercalation, only the hxylammonium cation with positive charge is intercalated into the layered $TiS_2$. As the hydrogen atoms are sensitive to the charge of the neighboring atoms, NMR spectrum of the hybrid material was examined and was found to be exactly the same as the reference pattern of pure hexylammonium chloride (Supplementary Fig. 4). Therefore, it can be concluded that only hexylammonium cations (protonated amine) are present in the hybrid material, which correspond to the electrons in the $TiS_2$ layers due to the electrical neutrality requirement.

The in-plane thermoelectric properties of the vacuum-heated $TiS_2(HA)_{0.025}$ were measured using the similar procedures reported in ref. 24 and the results are shown in Fig. 2. The thermoelectric properties of the $TiS_2$ single crystal and the previously reported $TiS_2(HA)_{0.08}(H_2O)_{0.22}(DMSO)_{0.03}$ were also included for comparison. The electrical conductivity of $TiS_2(HA)_{0.025}$ is comparable to that of the $TiS_2$ single crystal but much lower than that of $TiS_2(HA)_{0.08}(H_2O)_{0.22}(DMSO)_{0.03}$. The results of the Hall measurement were shown in Table 1. The carrier concentration of $TiS_2(HA)_{0.025}$ is slightly higher than that of the $TiS_2$ single crystal but much lower than $TiS_2(HA)_{0.08}(H_2O)_{0.22}(DMSO)_{0.03}$. The pristine $TiS_2$ single crystal is in fact a narrow band gap semiconductor and a tiny amount of Ti atoms can be self-intercalated into the van der Waals gap during the crystal growth process, introducing additional electrons[30]. $TiS_2(HA)_{0.08}(H_2O)_{0.22}(DMSO)_{0.03}$ was prepared by an electrochemical intercalation method, in which the $TiS_2$ layers were reduced and the electron density was equal to the density of the organic cations according to the electroneutrality requirement. When the density of organic cations is very high, the corresponding carrier concentration is also high, resulting in a low Seebeck coefficient and thus the low power factor. In $TiS_2(HA)_{0.025}$, most of the organic cations were evaporated, which significantly reduces the corresponding carrier concentration. The results confirm that the carrier concentration can be effectively tuned by the removal of organic cations.

Since electrons mainly move inside the inorganic $TiS_2$ layers, the mobility of the intercalated layered materials remain very high. However, they are all lower than the $TiS_2$ single crystal, as the intercalated organic cations can scatter electrons due to the electrostatic force[23]. In addition, it has been further confirmed that polar neutral solvent molecules, such as $H_2O$ and DMSO, surrounding the cations can reduce the electrostatic force by the dielectric screening effect. However, in

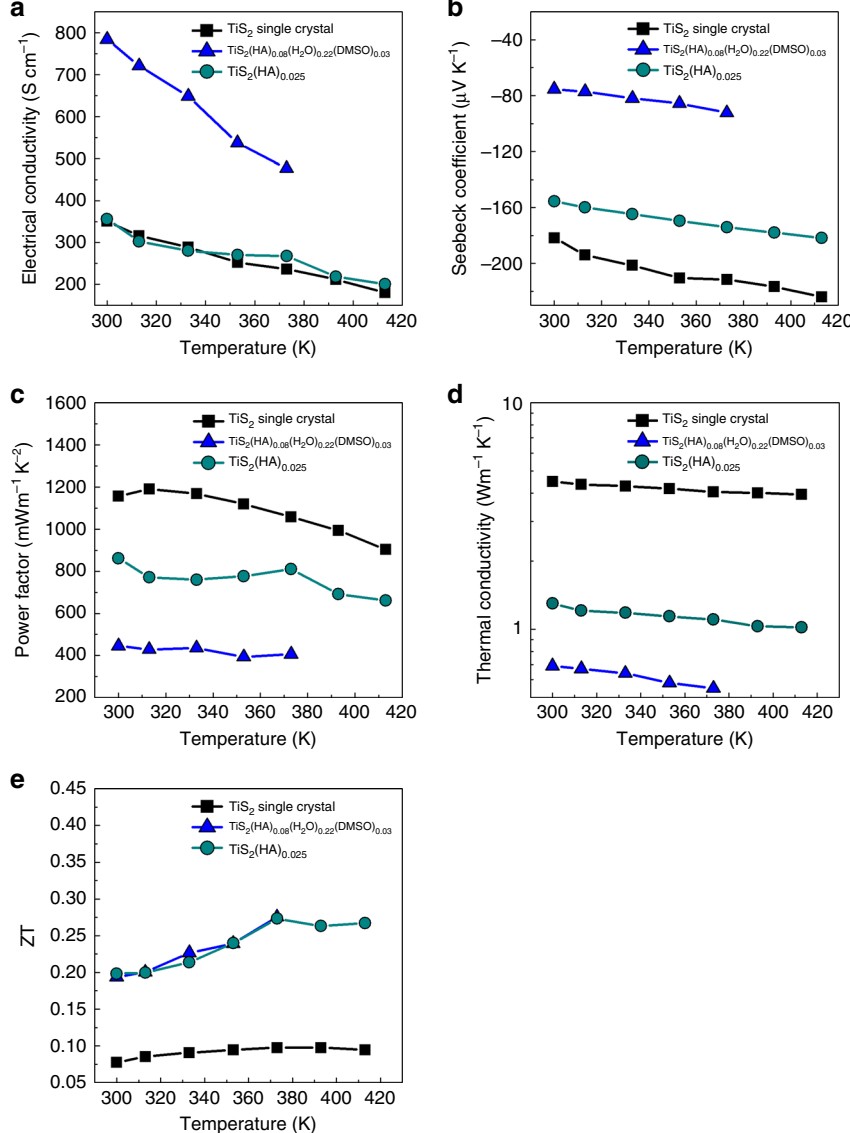

**Fig. 2** In-plane thermoelectric properties of $TiS_2(HA)_{0.25}$ compared with $TiS_2$ single crystal and $TiS_2(HA)_{0.08}(H_2O)_{0.22}(DMSO)_{0.03}$: **a** in-plane electrical conductivity, **b** in-plane Seebeck coefficient, **c** in-plane power factor, **d** in-plane thermal conductivity, **e** in-plane thermoelectric figure of merit, ZT

### Table 1 Carrier concentration and mobility of $TiS_2$ single crystal and the hybrid inorganic/organic superlattices

| Compositions | Carrier concentration (cm⁻³) | Mobility (cm² V⁻¹ S⁻¹) |
|---|---|---|
| $TiS_2$ single crystal | $1.8 \times 10^{20}$ | 11.4 |
| $TiS_2(HA)_{0.08}(H_2O)_{0.22}(DMSO)_{0.03}$ | $7.6 \times 10^{20}$ | 6.4 |
| $TiS_2(HA)_{0.025}$ | $4.0 \times 10^{20}$ | 5.4 |
| $TiS_2(TBA)_{0.025}(HA)_{0.012}$ | $5.0 \times 10^{20}$ | 5.0 |
| $TiS_2(TBA)_{0.013}(HA)_{0.019}$ | $4.8 \times 10^{20}$ | 5.8 |

Hall effect measurements of the carrier concentration and mobility

$TiS_2(HA)_{0.025}$, the polar organic molecules DMSO have been completely evaporated, so the mobility is lower compared with $TiS_2(HA)_{0.08}(H_2O)_{0.22}(DMSO)_{0.03}$, due to the absence of the dielectric screening effect by the polar organic molecules.

The Seebeck coefficient is consistent with the results of carrier concentration. The Seebeck coefficient of $TiS_2(HA)_{0.025}$ is lower than that of $TiS_2$ but is significantly higher than that of

$TiS_2(HA)_{0.08}(H_2O)_{0.22}(DMSO)_{0.03}$. As a result of the enhanced Seebeck coefficient, the power factor of $TiS_2(HA)_{0.025}$ is almost doubled compared with that of the previous $TiS_2(HA)_{0.08}(H_2O)_{0.22}(DMSO)_{0.03}$, which is among the best *n*-type flexible thermoelectric materials[3].

The in-plane thermal conductivities of the three materials were plotted in Fig. 2d. Apparently, the thermal conductivity in the hybrid materials is significantly reduced compared with that of the $TiS_2$ single crystal. As suggested by the molecular dynamics simulations, the heat-carrying phonons in the hybrid inorganic/organic superlattices are vigorously damped by the randomly dangling organic molecules[24]. However, the thermal conductivity of $TiS_2(HA)_{0.025}$ is much higher than that of $TiS_2(HA)_{0.08}(H_2O)_{0.22}(DMSO)_{0.03}$. This is because: (1) There are fewer organic molecules in $TiS_2(HA)_{0.025}$ than that in $TiS_2(HA)_{0.08}(H_2O)_{0.22}(DMSO)_{0.03}$. (2) The structure is randomly staged and organic molecules are intercalated in every two or three layers. It has been shown[24] that the extremely low thermal conductivity of the $TiS_2$/organic superlattice along the in-plane direction is due to organic molecules that are chemically bonded

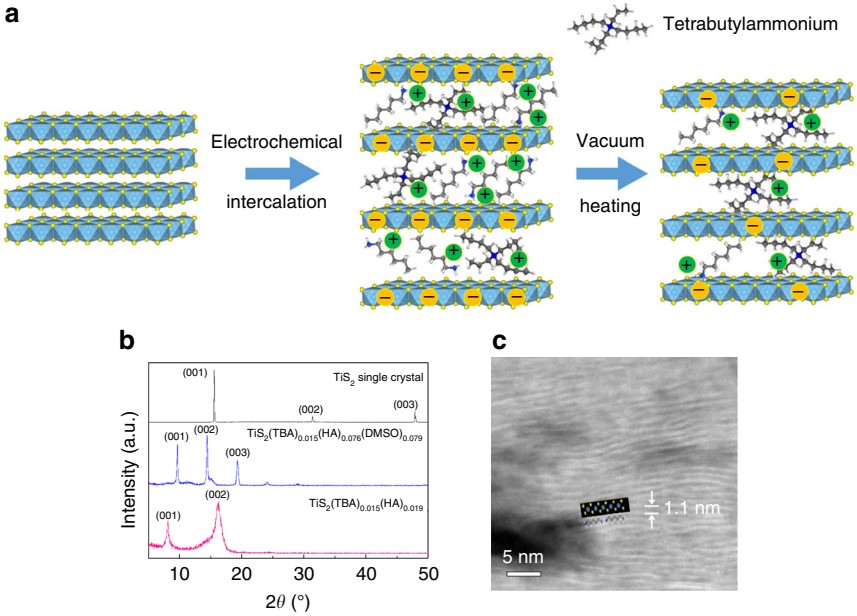

**Fig. 3 a** Evolution of structure and carrier concentration of $TiS_2$ single crystal with the electrochemical intercalation of HA/TBA molecules and vacuum heating. **b** XRD patterns of $TiS_2$ single crystal, $TiS_2(TBA)_{0.015}(HA)_{0.074}(DMSO)_{0.079}$ and $TiS_2(TBA)_{0.013}(HA)_{0.019}$. **c** Magnified HAADF-STEM image of $TiS_2(TBA)_{0.013}(HA)_{0.019}$

to the $TiS_2$ layers. Molecular dynamics simulations suggest that the organic molecules have random and independent vibrations that strongly disturb phonon transport inside the $TiS_2$ layers. In the high-stage structure, a large portion of the $TiS_2$ layers is not chemically bonded to the organic molecules, for which the disturbing effect of the organic molecules disappears. Therefore, the stage-1 structure is preferred to ensure that every possible $TiS_2$ layer is bonded to the organic molecules for a low thermal conductivity. This is consistent with a recent paper on intercalated graphite compound, which shows that the thermal conductivity of lithiated graphite compounds with a stage-2 structure ($LiC_{18}$) has a higher in-plane thermal conductivity than a stage-1 structure ($LiC_6$)[31].

As shown in Fig. 2e, ZT value of $TiS_2(HA)_{0.025}$ is comparable to that of $TiS_2(HA)_{0.08}(H_2O)_{0.22}(DMSO)_{0.03}$, but much larger than that of the $TiS_2$ single crystal. Even though the power factor is remarkably improved by reducing the carrier concentration when organic cations are evaporated, the increase of thermal conductivity prevents further improvement of the ZT value.

**Carrier optimization with dual-molecules intercalation.** The reduction of carrier concentration remarkably increases the power factor, but fails to improve the ZT value, as the thermal conductivity is increased as a result of formation of the random high-staging structure. The thermal conductivity can potentially be further reduced with small amount of organic molecules if the molecules are distributed uniformly inside the van der Waals gaps of the layered $TiS_2$ single crystal, forming a stage-1 structure. We have thus developed an innovative strategy to evaporate the organic molecules while maintaining the stage-1 structure. Here we co-intercalate two kinds of organic molecules with different atomic weight and boiling points. When heated in vacuum at an intermediate temperature between the two boiling points, the small molecules with a lower boiling point are evaporated, which leads to the reduction of carrier concentration according to the electroneutrality requirement. Meanwhile, the heavy molecules with a higher boiling point remain uniformly distributed in the $TiS_2$ van der Waals gaps and can pillar the layered structure to maintain the stage-1 structure. The great advantage of this

strategy is that the amount of evaporated low-boiling point molecules can be adjusted by changing the ratio between the two kinds of molecules, which results in great tunability of carrier concentration. (Supplementary Fig. 5)

The results reported above confirmed that large amounts of HA molecules can be evaporated by vacuum heating at 180 °C for 1 h. Therefore, HA was chosen as the low-boiling point component. Tetrabutylammonium (TBA) molecules with a high atomic weight and a high boiling point (>200 °C) was chosen as another component. In a preliminary experiment (Supplementary Fig. 6), pure TBA molecules were electrochemically intercalated and heated in vacuum at 180 °C for 1 h. The Seebeck coefficient was slightly changed from −59.6 to −63.8 μV K$^{-1}$, suggesting that most TBA molecules were maintained, resulting in a slight decrease of carrier concentration. Therefore, by intercalating a mixture of TBA and HA molecules into $TiS_2$ and heating the hybrid material at 180 °C for 1 h, most TBA molecules remained uniformly distributed in the van der Waals gaps as it was intercalated, with stage-1 final structure for the hybrid super-lattice. In addition, the ratio between the TBA and HA molecules can be tuned by the relative concentration of the two molecules in the electrolyte solution. The XRD patterns for the electrochemical intercalated products with different electrolyte solutions were shown in Supplementary Fig. 7. The interlayer distance for the products with different [TBA]/[HA] ratios were all around 1.6 nm and the patterns are similar, suggesting that the structure is similar. After vacuum heating, the interlayer distance decreases to 1.1 nm and the XRD patterns are also similar (Supplementary Fig. 8), except the [TBA]/[HA] = 1:10 case. Seebeck coefficients of the different compositions were measured and shown in Supplementary Fig. 9. With decreasing [TBA]/[HA] ratio (TBA mole fraction), the Seebeck coefficient increases gradually, suggesting a decrease in carrier concentration.

Samples with high Seebeck coefficients were selected, in which the initial ratio between TBA and HA molecules is 1:5 and 1:7 in the electrolyte solution, respectively. (The 1:10 sample was excluded, as the XRD pattern is close to the previous random stage $TiS_2(HA)_{0.025}$ sample in Fig. 1b, indicating the formation of a staging disorder structure). NMR was first used to analyze the

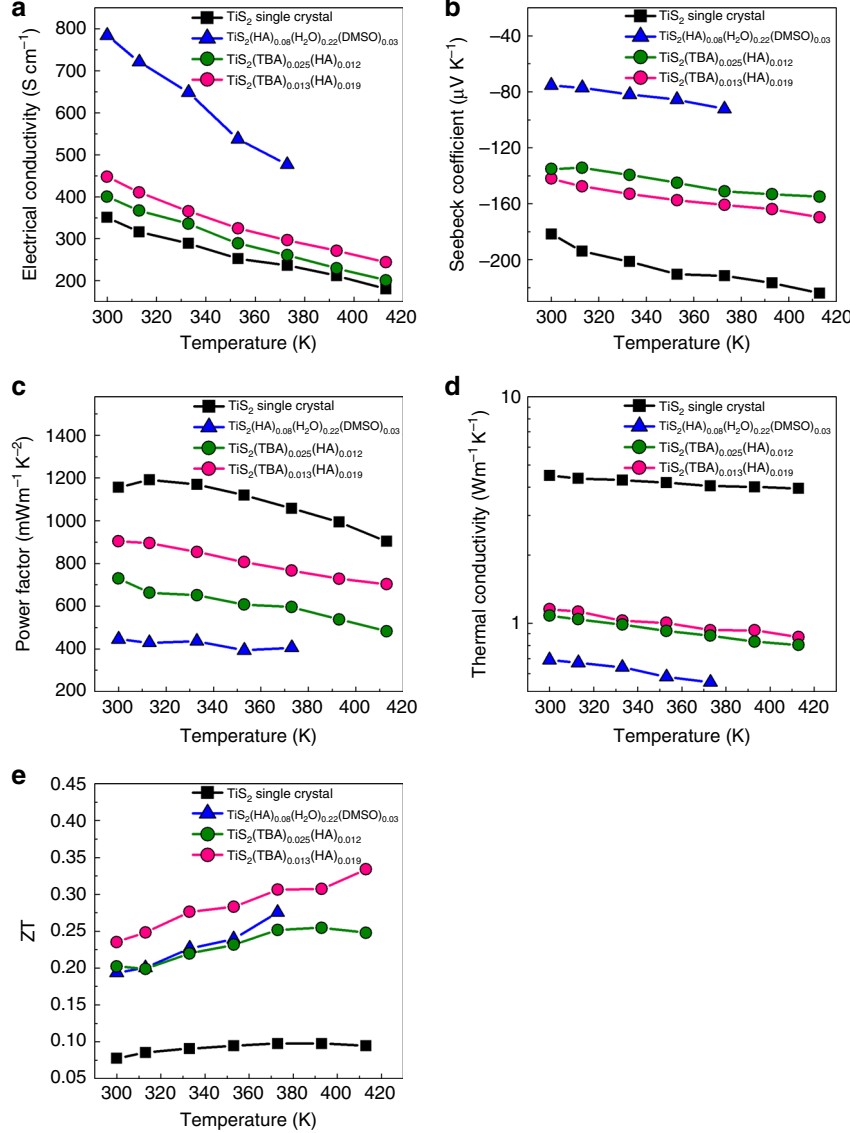

**Fig. 4** In-plane thermoelectric properties of $TiS_2(TBA)_{0.025}(HA)_{0.012}$, $TiS_2(TBA)_{0.013}(HA)_{0.019}$ compared with $TiS_2$ single crystal and $TiS_2(HA)_{0.08}(H_2O)_{0.22}(DMSO)_{0.03}$: **a** in-plane electrical conductivity, **b** in-plane Seebeck coefficient, **c** in-plane power factor, **d** in-plane thermal conductivity, **e** in-plane thermoelectric figure of merit, ZT

compositions. For the [TBA]:[HA] = 1:5 sample, the electrochemical intercalation gives a hybrid material with a composition of $TiS_2(TBA)_{0.026}(HA)_{0.054}(DMSO)_{0.034}$. The ratio between TBA and HA deviates from the original 1:5 in the electrolyte solution due to the different intercalation ability of the two molecules. After vacuum heating at 180 °C, the composition becomes $TiS_2(TBA)_{0.025}(HA)_{0.012}$. The neutral DMSO molecules were all evaporated as they were weakly bonded with the organic ions. A large portion of the lower boiling point HA molecules was also evaporated, resulting in a reduction of the number in the chemical formula. The heavy TBA molecules with a higher boiling point almost did not evaporate, and the change of the molecular proportion before and after vacuum heating remains within experimental errors. For the [TBA]:[HA] = 1:7 case, the composition of the electrochemical intercalated material was determined to be $TiS_2(TBA)_{0.015}(HA)_{0.074}(DMSO)_{0.079}$. After vacuum heating, the composition became $TiS_2(T-BA)_{0.013}(HA)_{0.019}$. The amount of the remaining molecules was much fewer than the [TBA]:[HA] = 1:5 case. The results also suggest the final compositions of the hybrid sample can be

effectively tuned by the initial ratio between the TBA and HA ions in the electrolyte solution, rendering the tunability of carrier concentration.

The HRTEM picture of the $TiS_2(TBA)_{0.013}(HA)_{0.019}$ sample, as shown in Fig. 3, clearly suggests a stage-1 layered structure, where the inorganic layers (bright area) and the organic layers stack alternatively. The interlayer distance is 1.1 nm, which is in good agreement with the XRD patterns. The result suggests that the stage-1 superlattice is well maintained by mixing TBA ions and the HA ions. The mixture of the two molecules is initially uniformly distributed in the interlayer space to form a stage-1 compound. When heated at a temperature between two boiling points of HA and TBA, most of the HA molecules are evaporated. However, the TBA molecules remain at their original positions as they were initially intercalated, because the temperature is not high enough to drive the TBA molecules to be mobile. Therefore, the stage-1 structure can be maintained. Furthermore, a hybrid inorganic–organic stage-1 $TiS_2(TBA)_x(HA)_y$ superlattice with tunable $x$ and $y$ has been synthesized by electrochemical intercalation of $TiS_2$ with mixed solution of $(TBA)_x(HA)_y$. By

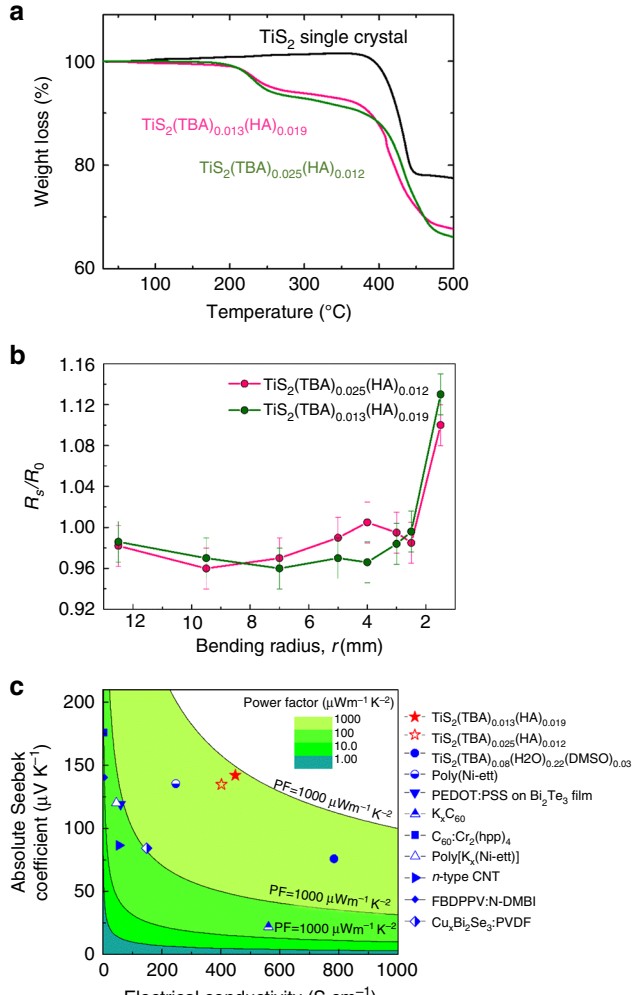

**Fig. 5 a** TG-DTA curve of $TiS_2$ single crystal, $TiS_2(TBA)_{0.025}(HA)_{0.012}$ and $TiS_2(TBA)_{0.013}(HA)_{0.019}$. **b** The sheet resistance $R$ as a function of bending radius for a $TiS_2(TBA)_{0.025}(HA)_{0.012}$ and $TiS_2(TBA)_{0.013}(HA)_{0.019}$ crystals, where $R_0$ is the corresponding value of its original state before bending. **c** Power factor of $TiS_2(TBA)_{0.025}(HA)_{0.012}$ and $TiS_2(TBA)_{0.013}(HA)_{0.019}$ compared with the other n-type flexible thermoelectric materials

selectively evaporating the lower boiling point organic cations, the carrier concentration in the hybrid superlattice can be easily tuned.

The in-plane thermoelectric properties were measured and shown in Fig. 4. Both the electrical conductivities of the $TiS_2(TBA)_{0.025}(HA)_{0.012}$ and $TiS_2(TBA)_{0.013}(HA)_{0.019}$ samples are lower than the previously reported $TiS_2(HA)_{0.08}$ $(H_2O)_{0.22}(DMSO)_{0.03}$ sample, but much higher than that of the $TiS_2$ single crystal. The results of the Hall measurements are shown in Table 1. The variation of carrier concentrations can account for the difference between the electrical conductivities. The results confirm that the evaporation of low-boiling point organic molecules have successfully decreased the carrier concentration, which is much lower than the $TiS_2(HA)_{0.08}(H_2O)_{0.22}(DMSO)_{0.03}$ sample. According to the electroneutrality principle, the total carrier concentration can be estimated from the chemical composition. Supposing that each organic cation corresponds to one elementary charge, the carrier concentration for the $TiS_2(TBA)_{0.025}(HA)_{0.012}$ and $TiS_2(TBA)_{0.013}(HA)_{0.019}$ sample can be estimated to be $3.3 \times 10^{20}$ cm$^{-3}$ and $2.8 \times 10^{20}$ cm$^{-3}$, respectively. By adding the

electron density $(1.8 \times 10^{20}$ cm$^{-3})$ inside the original $TiS_2$ single crystal due to the interstitial Ti atoms (Table 1), the final carrier concentration is determined to be $5.1 \times 10^{20}$ cm$^{-3}$ and $4.6 \times 10^{20}$ cm$^{-3}$, which is in reasonable agreement with the measured values. It also shows that by changing the initial ratio between TBA and HA molecules, the carrier concentration can be effectively tuned as expected.

The electron mobilities are all lower than the values of the $TiS_2$ single crystal and the previously reported $TiS_2(HA)_{0.08}$ $(H_2O)_{0.22}(DMSO)_{0.03}$. It is believed that organic cations in the hybrid superlattices can scatter electrons due to electrostatic force, which results in a reduction of electron mobility compared with the $TiS_2$ single crystal. Both $TiS_2(TBA)_{0.025}(HA)_{0.012}$ and $TiS_2(TBA)_{0.013}(HA)_{0.019}$ have much lower cation densities than that of $TiS_2(HA)_{0.08}(H_2O)_{0.22}(DMSO)_{0.03}$ and are therefore supposed to show higher electron mobility. However, as discussed earlier, the polar DMSO and $H_2O$ molecules in $TiS_2(HA)_{0.08}$ $(H_2O)_{0.22}(DMSO)_{0.03}$ can effectively screen the electrostatic potential of the organic cations, thereby restoring the electron mobility in the $TiS_2$ layers. In the current $TiS_2(TBA)_{0.025}(HA)_{0.012}$ and $TiS_2(TBA)_{0.013}(HA)_{0.019}$ compositions, polar DMSO molecules have been evaporated with the HA ions left. Therefore, the dielectric screening effect disappears and the electron mobility decreases compared with $TiS_2(HA)_{0.08}(H_2O)_{0.22}(DMSO)_{0.03}$.

The Seebeck coefficients of the $TiS_2(TBA)_{0.025}(HA)_{0.012}$ and $TiS_2(TBA)_{0.013}(HA)_{0.019}$ samples are all much higher than that of the $TiS_2(HA)_{0.08}(H_2O)_{0.22}(DMSO)_{0.03}$ due to the lower carrier concentration shown in Table 1. The Seebeck coefficient of the $TiS_2(TBA)_{0.013}(HA)_{0.019}$ sample is higher than the $TiS_2$ $(TBA)_{0.025}(HA)_{0.012}$ sample, also because of the lower carrier concentration. The power factors of the samples are shown in Fig. 4c. As predicted in the Supplementary Fig. 1, the power factor increases as a function of the reduced carrier concentration. The power factors of the $TiS_2(TBA)_{0.025}(HA)_{0.012}$ and $TiS_2$ $(TBA)_{0.013}(HA)_{0.019}$ samples are remarkably improved compared with that of the $TiS_2(HA)_{0.08}(H_2O)_{0.22}(DMSO)_{0.03}$ sample. For $TiS_2(TBA)_{0.025}(HA)_{0.012}$, the power factor reaches $904$ μW m$^{-1}$ K$^{-2}$ at 300 K, which is among the best n-type flexible thermoelectric materials. Only recently it has been realized that a high power factor is also very important for thermoelectric power generation[32]. This is more significant in organic thermoelectric materials than in inorganic thermoelectric materials because organic materials always have low thermal conductivity to maintain the temperature difference in a thermoelectric module in real applications. It is believed the main obstacle for organic materials is its low power factor compared with the inorganic materials, which is most critical for their power output[33].

The thermal conductivities of the $TiS_2(TBA)_{0.025}(HA)_{0.012}$ and $TiS_2(TBA)_{0.013}(HA)_{0.019}$ samples are all much lower than the $TiS_2$ single crystal (Fig. 4d). Using the Wiedemann–Franz law, the electronic thermal conductivities were calculated and the lattice thermal conductivities were obtained by substracting the electronic contribution from the total thermal conductivity. The lattice thermal conductivities were estimated to be 0.92 and 0.87 W m$^{-1}$ K$^{-1}$ for the $TiS_2(TBA)_{0.025}(HA)_{0.012}$ and $TiS_2(TBA)_{0.013}$ $(HA)_{0.019}$ samples, respectively, which are significantly lower than the pristine $TiS_2$ samples (4.5 W m$^{-1}$ K$^{-1}$). Molecular dynamic simulations have clarified the mechanism of the thermal conductivity[24]. It has been found that the acoustic phonons, especially transverse acoustic phonons were significantly scattered by the organic molecules that are chemically bonded to the $TiS_2$ layers, leading to a large reduction of phonon mean free path and thermal conductivity. Meanwhile, it is also found that the reduction of thermal conductivity is not as huge as the $TiS_2(HA)_{0.08}(H_2O)_{0.22}(DMSO)_{0.03}$ sample[24]. It is related with the reduction in the amount of organic

molecules. In $TiS_2(HA)_{0.08}(H_2O)_{0.22}(DMSO)_{0.03}$, the total amount of organic molecules is about 0.33 mole per unit cell of $TiS_2$. The value becomes 0.037 and 0.032 for the $TiS_2(TBA)_{0.025}(HA)_{0.012}$ and $TiS_2(TBA)_{0.013}(HA)_{0.019}$ samples, which are ten times lower than that in $TiS_2(HA)_{0.08}(H_2O)_{0.22}(DMSO)_{0.03}$. Therefore, the phonon scattering strength is weakened and the lattice thermal conductivity is not as low as the previous $TiS_2(HA)_{0.08}(H_2O)_{0.22}(DMSO)_{0.03}$ sample.

Due to the large improvement of the power factor, the ZT value of the $TiS_2(TBA)_{0.025}(HA)_{0.012}$ sample was improved to be 0.33 at 413 K, which is among the highest in the *n*-type flexible thermoelectric materials. It shows great potential in flexible thermoelectric modules as a counterpart of the *p*-type conducting polymers, such as PEDOT-PSS.

Thermal stability of the inorganic/organic superlattices were examined using the TG-DTA measurement as shown in Fig. 5a. These two materials are stable until 175 °C in air atmosphere without detectable mass loss. It suggests that the organic molecules that are spatially confined in the van der Waals gap of $TiS_2$ layers can have better thermal stability than their ordinary liquid state. Therefore, energy harvesting and Peltier cooling around room temperature for the hybrid material can be guaranteed.

The mechanical flexibility of the hybrid inorganic/organic superlattices was further measured and shown in Fig. 5b. The materials were attached to the surface of glass tube with different radii. The electrical resistance was measured as a function of the bending radius. It can be found the resistance of both materials can be maintained within 5% of its original state until a bending radius of 2.5 mm, suggesting an excellent flexibility. The interlayer expansion by the soft organic molecules can account for the flexibility.

## Discussion
To summarize, we developed a strategy to tune the carrier concentration of the hybrid inorganic/organic superlattices without breaking the layer-by-layer structure. We electrochemically intercalate two kinds of organic materials TBA and TA molecules with different boiling points into the van der Waals gap between the inorganic layers. By vacuum heating, the materials at an intermediate temperature between the two boiling points, HA molecules were evaporated which reduces the carrier density in the $TiS_2$ layers due to the electroneutrality requirement. The stage-1 structure is remained with the high boiling point TBA molecules uniformly distributed in the van der Waals gaps. The carrier concentrations can be effectively reduced by 2–3 times. A remarkably high power factor of 904 $\mu$W m$^{-1}$ K$^{-2}$ was obtained, which is very high among the recently developed *n*-type flexible thermoelectric materials (Fig. 5c) and even approaches the level of the inorganic materials. Together with a high thermal stability and good flexibility, the $TiS_2$-based inorganic/organic superlattices have shown great promise in flexible modules for wearable energy harvesting or personal temperature management.

## Methods
**Synthesis**. $TiS_2$ single crystals with a typical size of 4 mm × 4 mm × 100 μm were fabricated by chemical vapor transport method[30]. Organic molecules were electrochemically intercalated into the $TiS_2$ crystals, as shown in Supplementary Fig. 2. The $TiS_2$ single crystals and platinum plate were used as cathode and anode, respectively. TBA and HA chloride dissolved into DMSO was used as electrolyte. The electrochemical intercalation was performed under a constant voltage of 1.5 V. The obtained hybrid inorganic–organic materials were then vacuum heated to selectively evaporate the organic molecules.

**Characterization and measurement**. The structure was analyzed using XRD and HAADF-STEM. The composition was quantitatively analyzed by $^1$H NMR using dimethyl sulfone as a reference material. The thermal stability was measured using a TG-DTA system (TG8120, Rigaku). The hall measurement was performed in a commercial system (ResiTest8300, Rigaku). The electrical conductivity and Seebeck coefficients were measured using a home-made apparatus, which had been well calibrated. The thermal diffusivity was measured using laser flash method as introduced in Supplementary Fig. 10. The heat capacity was measured using DSC method and the thermal conductivity was calculated as a product of thermal diffusivity, heat capacity, and density. The resistance as a function of bending radius was described in Supplementary Fig. 11.

**Data availability**. The data that support the findings of this study are available from the corresponding author on request.

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

## Acknowledgements

C.W. acknowledges financial support from 1000 Plan Program for Young Talents of China. K.K. acknowledges financial support from JSPS KAKENHI Grant no. 25289226 and NEDO-TherMat. R.Y. acknowledges the support from the US National Science Foundation (Grant No. 1512776).

## Author contributions

C.W. initiated the concepts and designed the experiments. C.W., R.T., M.C., and P.Z. conducted the experiments, including the synthesis, characterization, and the measurements. C.W. and R.Y. did the analysis and wrote the manuscript. All of the authors contributed to manuscript preparation.

## Additional information

**Competing interests:** The authors declare no competing financial interests.

