## [Peer Review File · Nature Communications]

Reviewers' comments:

Reviewer #1 (Remarks to the Author):

This is an interesting manuscript in which the use of organic intercalation to vary the thermoelectric properties of TiS_2 is reported. The approach of introducing two amines with quite different molecular weights and then selectively removing (most of) the lighter one by an appropriate thermal treatment in order to produce a stage 1 phase is ingenious. It is a little surprising that the ordered stage 1 intercalate exhibits a lower thermal conductivity than that of a random high-stage structure. Disorder is usually beneficial in achieving reductions in thermal conductivity. Perhaps the authors could comment on the origin of this effect. They also need to explain more clearly the advantages of this approach over one in which the heavier amine alone is intercalated, as there seems to be a fairly limited tunability of the properties. It is unclear to what extent control can be exerted over the properties. At higher TBA:HA ratios, powder X-ray diffraction appears to indicate the formation of additional phases in the initially-formed products (Figure S6), which may restrict the effective range over which tunability may be exerted.

Implicit in the discussion of the results of the transport properties is a one to one correspondence between the amount of amine incorporated and the number of electrons transferred. There are reports that suggest the intercalation of an organic amine into a dichalcogenide is accompanied by side reactions which lead to the presence of both neutral and cationic species in the inter-layer space. How has the presence of only protonated amines been confirmed?

The manuscript represents an extension of the authors' initial report (Ref. 21) of the impact of electrochemical intercalation of hexylammonium on the thermoelectric properties of TiS_2 . The figure of merit is increased only very slightly from that of the earlier work and the difference in the values (0.33 vs. 0.28) appears to be close to the limits of experimental uncertainty. This very similar thermoelectric performance lessens the impact of the work, despite the innovative approach that has been adopted. For this reason, I believe that the manuscript falls just short of what would be expected for publication in Nature Communications.

Reviewer #2 (Remarks to the Author):

Ultrahigh thermoelectric power factor in n-type flexible hybrid inorganic-organic superlattice

The paper communicates strategies to tune the carrier concentration of the hybrid inorganic/organic superlattices using electrodeposition method. Up to two organic materials were intercalated into layers of TiS_2 films. The vacuum heating then resulted in n-type thermoelectric hybrid materials with high ZT value. The samples are flexible and sort of solution-processed, at least partially, in line with low-cost fabrication of thermoelectric devices. The analyses, results, and discussions are sound and systematic and the developed strategy is novel and the hybrid material is of high performance. Several parts need some clarifications and improvement including the "introduction" and the "synthesis process of the samples". There are numerous typos as well, which need to be fixed. The paper is recommended for publication, after the following points are addressed.

1. Line 46 to 55: Thin film thermoelectrics: When authors discuss about the flexible wearable thermoelectric materials, in fact, they are referring to flexible thin film thermoelectric, compared to bulk thermoelectric materials. Therefore, it is constructive to mention the term "thin film thermoelectric" as well. Some of the cited references, in fact, deal with thin film thermoelectric materials, but the paper does not clarify it directly. The following reference discusses the differences between bulk and thin film thermoelectric devices and the advantages of thin film

thermoelectrics, which I believe should be mentioned here to clarify at the beginning that whether the paper considers bulk or thin film thermoelectrics. Section 6 of this paper: Eslamian, M. Nano-Micro Lett. (2017) 9: 3. doi:10.1007/s40820-016-0106-4

2. Line 56 to 59: Related to comment 1, as well. In addition to the requirement for flexibility, it is recommended to revise the discussion of these lines based on the condition of solution-processibility of different materials, as well, a topic also discussed in the reference mentioned in comment 1. In other words, both flexibility and solution-processibility matter, and organic materials and organic composites have both advantages.

3. Lines 60 to 65: Authors state that: "It is very difficult to make high performance n-type organic..." It will be constructive to mention some pristine n-type organic and their corresponding power factor or ZT to back the argument.

4. Line 96- 97. Please provide the value of the power factor and reference for the inorganic counterpart.... "An extremely large power factor ($904\mu\text{W}/\text{mK}^2$) was obtained for the n-type flexible material at room temperature, which is very close to that of the high efficiency inorganic thermoelectric materials."

5. Line 362: Starting from this line, the layer by layer fabrication process is not clear. The electrodeposition starts with TiS_2 thin film or plate inserted into the electrolyte and the organic materials....Then how the layer by layer fabrication process of the hybrid thermoelectric film proceeds? Vertical layers can be confused with horizontal layers of this work. Details of the fabrication process must be revealed so that others could reproduce the results. Figure 1a, sort of depicts what the authors mean by "layer by layer fabrication", but it should be mentioned in the texts to avoid confusion. It is recommended that the authors briefly explain or at least mention how CVD results in that kind of structure for TiS_2 , as shown in Fig. 1a.

6. Line 118: Authors state that: "The Seebeck coefficient increases with increasing temperature, suggesting an increase of carrier concentration." This is however, against the results of Fig. S1 that show an inverse relationship between the Seebeck coefficient and the charge carrier density.

7. Line 340: "The materials were attached to the surface..." What are the dimensions of the samples? Are they in the category of thin films?

8. Minor errors and typos:

- Line 50-51: Especially, these flexible thermoelectric materials "is" now under active pursuit for wearable energy harvesting, as "it" can...
- Line 86: We have recently "demonstrate" a hybrid...
- Line 106: anode where the negative charges "compensates" the intercalated
- Line 112: The first attempt was "tried"...
- Line 122: Then the sample was heated at 180°C for different "time"...
- Line 124: then "saturate" even with a duration as long as 3 hours...
- Line 207: Meanwhile, the remaining heavy molecules with a higher "boiling" "remains" uniformly distributed in the TiS_2 van...
- Line 283: "...The electron "motilities" are all lower than..."
- Line 332: ... PSS-PEDOT"...should be PEDOT:PSS...
- Line 348: "We electrochemically intercalate two kinds of organic materials TBA and TA molecules with different boiling points into the "organic" layers.."
- Line 373: Some typos: "The thermal stability was measured using a TG-DTA system. (Rigaku, 374 TG8120) The hall measurement was done in a commercial system ((Toyo, ResiTest8300))."
- Line 363: $4\text{mm} \times 4\text{mm} \times 100\mu\text{m}$ to be replaced with \times .

Response to Reviewer 1

Comment_0: *This is an interesting manuscript in which the use of organic intercalation to vary the thermoelectric properties of TiS₂ is reported. The approach of introducing two amines with quite different molecular weights and then selectively removing (most of) the lighter one by an appropriate thermal treatment in order to produce a stage 1 phase is ingenious.*

Response_0: Thank you for your supporting comment!

Comment_1: *It is a little surprising that the ordered stage 1 intercalate exhibits a lower thermal conductivity than that of a random high-stage structure. Disorder is usually beneficial in achieving reductions in thermal conductivity. Perhaps the authors could comment on the origin of this effect.*

Response_1: Thank you for your comment. In our previous paper [Wan, C., Gu, X., et al., *Nat. Mater.***14**, 622-625(2015)], we have demonstrated the extremely low thermal conductivity of the TiS₂/organic superlattice along the in-plane direction is due to the organic molecules that are chemically bonded to the TiS₂ layers. Using molecular dynamics simulations, we showed that the organic molecules have random and independent vibrations that strongly disturb phonon transport inside the TiS₂ layers. In the high-stage structure, a large portion of the TiS₂ layers are not chemically bonded to the organic layers, for which the disturbing effect of the organic molecules disappears. Therefore, stage-1 structure is preferred to reach a low thermal conductivity since every TiS₂ layer is bonded to organic molecules. We also note that it is not always true that a random structure would have a lower thermal conductivity than that of an ordered structure. In a recent paper on the correlation of staging and thermal conductivity of lithiated graphite compounds led by Yang, one of the coauthors of this manuscript, they also demonstrated that thermal conductivity in a random LiC₁₈ (stage-2' structure) has a higher in-plane thermal conductivity than LiC₆ (stage-1 structure) [Qian, X., Yang, R., et al, *J. Phys. Chem. Lett.*, **7**, 4744–4750(2016)]

Revision_1: Page 10, line 11-22. We have added the above explanation into the manuscript.

Comment_2: *They also need to explain more clearly the advantages of this approach over one in which the heavier amine alone is intercalated, as there seems to be a fairly limited tunability of the properties. It is unclear to what extent control can be exerted over the properties. At higher TBA:HA ratios, powder X-ray diffraction appears to indicate the formation of additional phases in the initially-formed products (Figure S6), which may restrict the effective range over which tunability may be exerted.*

Response_2: Thanks for this comment, which has indeed inspired us for further discussions.

First of all, the approach of single molecules intercalation will inevitably lead to

high stage layered compound when heated, no matter whether the molecules are light or heavy. According to Prof. M.S. Dresselhaus' review paper on intercalation compounds [page 32, Dresselhaus, M.S., Dresselhaus, G., *Advances in Physics*, **51**, 1-186(2002)], the molecules intercalated into a layered compound increases the strain energy by expanding the interlayer distance. When there are too few molecules to fill all the possible interlayer space, these molecules tend to aggregate to form clustering between a single set of the layers, rather than randomly and loosely distribute in all the interlayer space, because this aggregation can reduce the strain energy of the whole system. This explains why a random high stage compound is formed when $\text{TiS}_2[\text{HA}_x]$ is heated in vacuum, as seen in Figure 1c. Similarly, when we heat $\text{TiS}_2[\text{TBA}_x]$ above the boiling point of TBA, a random high stage compound will also be formed to minimize the strain energy. As we explained when addressing comment_1, a high stage compound does not necessarily show lower thermal conductivity.

The key innovation of the proposed approach of dual molecule intercalation in this work is that the mixture of the two molecules are initially uniformly distributed in the interlayer space to form a stage-1 compound. When heated at a temperature between two boiling points of HA and TBA, most of the HA molecules are removed. However, the TBA molecules remain at their original positions as they were initially intercalated, because the temperature is not high enough to drive the TBA molecules to be mobile. In this way, we were able to maintain the uniform stage-1 structure.

Regarding the tunability of the properties, we can vary the ratio, $[\text{TBA}]/[\text{HA}]$, to tune the carrier concentration. Theoretically, the ratio $[\text{TBA}]/[\text{HA}]$ can be changed from positive infinity to negative infinity with very large tunability of carrier concentration. However, in this paper, we only tried very small ratios from 1:1 to 1:7 where Seebeck coefficient varies from -115 to -142 $\mu\text{V}/\text{K}$, because our theoretical prediction shows that a high Seebeck coefficient is preferred for the case of TiS_2 intercalated compound. (The 1:10 composition is excluded here, as the content of TBA is too few to support the stage-1 structure after evaporation of the HA molecules, which is confirmed in Fig. S7). If we try large $[\text{TBA}]/[\text{HA}]$ ratios, from 10:1 to 1:1, only a few of HA molecules will be removed and the carrier concentration will still remain very high. The Seebeck coefficient will be lower than those with smaller $[\text{TBA}]/[\text{HA}]$ ratios. The extreme case is $\text{TiS}_2[\text{TBA}]_x$, which shows a Seebeck coefficient of -63.8 $\mu\text{V}/\text{K}$. In other words, by varying the $[\text{TBA}]/[\text{HA}]$ ratio, the Seebeck coefficient can be tuned from -63.8 $\mu\text{V}/\text{K}$ to -142 $\mu\text{V}/\text{K}$, which is truly a very wide range for the optimization of thermoelectric properties.

We believe that the approach of dual molecule intercalation presented here can be considered as a general methodology for tuning carrier concentration of hybrid inorganic/organic superlattices, since the conventional chemical doping is not applicable. There are also other opportunities, for example, using another combination of organic molecules, which have different geometry and molecular length that can better maintain uniform stage-1 layered structure.

Revision_2: Page 7, Line 11-19. The above discussion is added into the manuscript.

In the supplementary information, we added more description of tunability range into the caption of Fig. S9

Comment_3: *Implicit in the discussion of the results of the transport properties is a one to one correspondence between the amount of amine incorporated and the number of electrons transferred. There are reports that suggest the intercalation of an organic amine into a dichalcogenide is accompanied by side reactions which lead to the presence of both neutral and cationic species in the inter-layer space. How has the presence of only protonated amines been confirmed?*

Response_3: It is true that there are reports that both neutral amine and protonated amine coexist in the interlayer space, as there is an equilibrium between neutral amine and protonated amine in an aqueous solution [Schollho, R., Weiss, A., J. *Less-Common Metals*, **36**, 229-236(1974)]. However, in our experiment, we used an electrochemical intercalation method, which is quite different from the previous reports, such as the ion exchange method. The organic molecules to be intercalated are dissolved into solvent to form an electrolyte solution and the cations in the electrolyte are intercalated due to the electrostatic force. As a result, cationic amines will be intercalated. Meanwhile, we used hexylammonium chloride and tetrabutylammonium bromide as the solute molecules and dimethyl sulfoxide (DMSO) as a solvent. The solute molecules are all protonated amines and will not change to neutral amines in DMSO, which is a polar aprotic solvent. Therefore, in the electrolyte solution, there are no neutral amines at all. Even during the electrochemical intercalation process, the voltage we used is around -1.0~-1.5 V, which is far below the value that can reduce the protonated amines to neutral amines. Therefore, we believe that there are only protonated amines in the interlayer space.

Figure R1 ^1H NMR spectrum of (a) $\text{TiS}_2[\text{HA}_x]$ and (b) $\text{NH}_4^+\cdot\text{HCl}$ dissolved in D_2O .

To further confirm this, we compared the NMR spectrum of the hybrid sample with the reference spectrum, as shown below in Figure R1. As is known, the hydrogen

atoms are sensitive to the charge of the neighboring atoms, so neutral amines and protonated amines should have different spectrum. Fig. R1(a) shows the ^1H NMR spectrum of our sample, $\text{TiS}_2[\text{HA}_x]$, where the four peaks corresponding to different hydrogen atoms in hexylammonium (the protonated amines) can be seen. Fig. R1(b) shows a reference pattern for pure hexylammonium chloride (the protonated amines). [page S42 of the supporting material, Wittenberg, J.B., Zavalij, P. Y., et al. *Angew. Chem. Int. Edit.*, 52, 3690-3694(2013)]. We can find the peaks related to hexylammonium are exactly the same. Therefore, we can conclude that only protonated amines are present in the interlayer space.

Response_3: Page 8, Line 1-11. We added the discussion demonstrating that only protonated amines are present in the interlayer space. We also added the new figure S4 in the supplementary information.

Comment_4: *The manuscript represents an extension of the authors' initial report (Ref. 21) of the impact of electrochemical intercalation of hexylammonium on the thermoelectric properties of TiS_2 . The figure of merit is increased only very slightly from that of the earlier work and the difference in the values (0.33 vs. 0.28) appears to be close to the limits of experimental uncertainty. This very similar thermoelectric performance lessens the impact of the work, despite the innovative approach that has been adopted. For this reason, I believe that the manuscript falls just short what would be expected for publication in Nature Communications.*

Response_4: Even though the ZT value does not significantly change from our previous published paper in Nature Materials, we have significantly increased the power factor from $450 \mu\text{W}/\text{mK}^2$ to $904 \mu\text{W}/\text{mK}^2$. The doubling of power factor mean twice the power output. We would like to note that only very recently the research community begins to realize that power factor is equally important as ZT value for thermoelectric power generation. [(Liu, W., Kim, H.S., Jie, Q., Ren, Z., *Scripta Mater.*, 111, 3(2016))]. This is more significant for organic thermoelectric materials than in inorganic thermoelectric materials because organic materials usually have low thermal conductivity to maintain the temperature difference in a thermoelectric module in real applications. The main obstacle for organic materials is its low power factor compared with the inorganic materials, which is most critical for their power output. [Russ, B., Glauddell, A., Urban, J. et al, *Nat. Rev. Mater.*, 1, 1(2016)] Therefore, many works related to organic thermoelectric materials aim at improving the power factor. This is indeed why we titled this manuscript as “Ultrahigh thermoelectric power factor in n-type flexible hybrid inorganic-organic superlattice”.

Until now, the highest power factor for n-type organic materials is $453 \mu\text{W}/\text{mK}^2$. In this paper, due to an innovative doping optimization approach we proposed, we are able to tune the carrier concentration and significantly increase the thermoelectric power factor to $904 \mu\text{W}/\text{mK}^2$. In fact, this value is now comparable to high efficiency inorganic materials. For example, it is even close to that of SnSe ($1010 \mu\text{W}/\text{mK}^2 @ 850\text{K}$), an inorganic material with the best thermoelectric performance until now [Zhao, L.D., Lo, S.H., et al, *Nature*, 508, 373(2014)]. Therefore, we believe that the

power factor achieved in this hybrid material is truly a breakthrough.

Finally, we believe the approach of dual molecule intercalation we proposed in this work is a general methodology for tuning carrier concentration of hybrid inorganic/organic superlattices. It brings new opportunities, for example, by using another combination of organic molecules with different geometry and heavier molecular weight, which may further reduce thermal conductivity and enhance the thermoelectric performance.

Response_4: Page 16, line 11-17. We added the above discussion to emphasize the importance of thermoelectric power factor.

Response to Reviewer 2

Comment_0: *The paper communicates strategies to tune the carrier concentration of the hybrid inorganic/organic superlattices using electrodeposition method. Up to two organic materials were intercalated into layers of TiS₂ films. The vacuum heating then resulted in n-type thermoelectric hybrid materials with high ZT value. The samples are flexible and sort of solution-processed, at least partially, in line with low-cost fabrication of thermoelectric devices. The analyses, results, and discussions are sound and systematic and the developed strategy is novel and the hybrid material is of high performance. Several parts need some clarifications and improvement including the “introduction” and the “synthesis process of the samples”. There are numerous typos as well, which need to be fixed. **The paper is recommended for publication**, after the following points are addressed.*

Response_0: Thank you for your positive comments!

Comment_1: *Line 46 to 55: Thin film thermoelectrics: When authors discuss about the flexible wearable thermoelectric materials, in fact, they are referring to flexible thin film thermoelectric, compared to bulk thermoelectric materials. Therefore, it is constructive to mention the term “thin film thermoelectric” as well. Some of the cited references, in fact, deal with thin film thermoelectric materials, but the paper does not clarify it directly. The following reference discusses the differences between bulk and thin film thermoelectric devices and the advantages of thin film thermoelectrics, which I believe should be mentioned here to clarify at the beginning that whether the paper considers bulk or thin film thermoelectrics. Section 6 of this paper: Eslamian, M. Nano-Micro Lett. (2017) 9: 3. doi:10.1007/s40820-016-0106-4*

Response_1: Thanks for pointing this out. It is true that these flexible thermoelectric materials are mainly thin film thermoelectrics.

Revision_1: Page3, Line 1 and 5, we have adopted the term flexible thermoelectric thin film as marked in red color. The suggested reference is also included at the very beginning of the paper. (Reference 1)

Comment_2: *Line 56 to 59: Related to comment 1, as well. In addition to the requirement for flexibility, it is recommended to revise the discussion of these lines based on the condition of solution-processibility of different materials, as well, a topic*

also discussed in the reference mentioned in comment 1. In other words, both flexibility and solution-processibility matter, and organic materials and organic composites have both advantages.

Response_2: Thank you for a constructive comment. In page 3, line 13-15, the discussion is revised to be “In contrast, conducting polymers and polymer composites are much more advantageous to make desired flexible thermoelectric devices, which combines structural flexibility and solution-processibility” The reference mentioned in comment 1 is cited again.

Comment_3: Lines 60 to 65: Authors state that: “It is very difficult to make high performance n-type organic...” It will be constructive to mention some pristine n-type organic and their corresponding power factor or ZT to back the argument.

Response & Revision: Bottom lines of page3, and top lines of page 4. We added a sentence showing the power factor of two representative n-type polymers. “For example, a low power factor of $0.6\mu\text{W}/\text{mK}^2$ was reported for the n-type polymer poly[N,N'-bis(2-octyl-dodecyl)-1,4,5,8-naphthalenedicarboximide-2,6-diyl]-alt-5,5'-(2,2'-bithiophene)] (P(NDIOD-T2))¹⁰ and $1.4\mu\text{W}/\text{mK}^2$ was reported for the self-doped perylene diimides (PDI).¹¹”

Comment_4: Line 96- 97. Please provide the value of the power factor and reference for the inorganic counterpart.... “An extremely large power factor ($904\mu\text{W}/\text{mK}^2$) was obtained for the n-type flexible material at room temperature, which is very close to that of the high efficiency inorganic thermoelectric materials.”

Response and Revision: Page 5, Line 11-12. We added a sentence: “which is even close to that of some high-ZT inorganic materials, such as SnSe ($1010\mu\text{W}/\text{mK}^2$ @ 850K)²⁵ and Cu_{2-x}Se ($1200\mu\text{W}/\text{mK}^2$ @ 1000K)²⁶.”

Comment_5: Line 362: Starting from this line, the layer by layer fabrication process is not clear. The electrodeposition starts with TiS_2 thin film or plate inserted into the electrolyte and the organic materials....Then how the layer by layer fabrication process of the hybrid thermoelectric film proceeds? Vertical layers can be confused with horizontal layers of this work. Details of the fabrication process must be revealed so that others could reproduce the results. Figure 1a, sort of depicts what the authors mean by “layer by layer fabrication”, but it should be mentioned in the texts to avoid confusion. It is recommended that the authors briefly explain or at least mention how CVD results in that kind of structure for TiS_2 , as shown in Fig. 1a.

Response_5: Actually, Fig. 1a shows the intrinsic structure of 1T- TiS_2 , which has a CdI_2 structure with space group P-3m1. It is a quasi-2-dimensional structure, where the layers was connected by van der Waals gap. The TiS_2 single crystal was obtained by a standard chemical vapor transport (CVT) method. As shown in the revised Figure S2, titanium and sulfur powder mixture was sealed in evacuated silica tube. The silica tube was horizontally placed with the powders put at one end. Then the tube was put into a three-zone furnace, in which a temperature gradient across the tube was

established. By multiple tries, the best temperature for the hot end and cold end are 732 °C and 632 °C respectively. Additional sulfur was added as the agent which carries the vapor of Ti and Sulfur from the hot end to the cold end, where the single crystal of TiS_2 was obtained.

To fabricate the TiS_2 /organic layered material, an electrochemical intercalation process was used, which is very similar to the electrochemical reaction in lithium ion battery electrodes. During the charging process of a lithium ion battery, the negative electrode (graphite) was reduced and negatively charged, the positive lithium ions are intercalated into each van der Waals gap of graphite. In the revised Figure S2, we plotted a new figure to show the fabrication process of the TiS_2 /organic layered material. When the voltage is applied, the TiS_2 layers were negatively charged and the positive organic cations were intercalated into the van der Waals gap driven by electrostatic force, forming a layer by layer structure. In the real electrochemical reaction cell, the TiS_2 crystal was vertically placed. However, in the figures of the paper, the TiS_2 layers were horizontally placed for more convenient illustration of the structure. We are sorry for the confusion.

Figure R2 Illustration of the fabrication process of the TiS_2 crystal and TiS_2 /organic superlattice by chemical vapor transport method and electrochemical intercalation method.

Revision_2: We have substantially revised the supplementary Fig. S2 and its caption to

show the details of the fabrication process.

Comment_6: Line 118: Authors state that: “The Seebeck coefficient increases with increasing temperature, suggesting an increase of carrier concentration.” This is however, against the results of Fig. S1 that show an inverse relationship between the Seebeck coefficient and the charge carrier density.

Response_6: We apologize for our mistake. In Figure S3a, we increase the heating temperature and more organic cations are evaporated and the carrier concentration is reduced due to the electrical neutrality requirement between organic cations and electrons. Therefore, with increasing temperature, the carrier concentration is actually decreased, which would account for the increase in Seebeck coefficient.

Revision_6: Page 6, line 11. We changed this sentence as: “The Seebeck coefficient increases with increasing temperature, suggesting a decrease in carrier concentration.”

Comment_7: Line 340: “The materials were attached to the surface...” What are the dimensions of the samples? Are they in the category of thin films?

Response_7: The dimension is about 4mm × 4mm × 150μm and can be categorized as a thick film.

Comment_8: Minor errors and typos:

- Line 50-51: Especially, these flexible thermoelectric materials “is” now under active pursuit for wearable energy harvesting, as “it” can...
- Line 86: We have recently “demonstrate” a hybrid...
- Line 106: anode where the negative charges “compensates” the intercalated
- Line 112: The first attempt was “tried”...
- Line 122: Then the sample was heated at 180oC for different “time”...
- Line 124: then “saturate” even with a duration as long as 3 hours...
- Line 207: Meanwhile, the remaining heavy molecules with a higher “boiling” “remains” uniformly distributed in the TiS₂ van...
- Line 283: “...The electron “motilities” are all lower than...”
- Line 332: ... PSS-PEDOT” ...should be PEDOT:PSS...
- Line 348: “We electrochemically intercalate two kinds of organic materials TBA and TA molecules with different boiling points into the “organic” layers..”
- Line 373: Some typos: “The thermal stability was measured using a TG-DTA system. (Rigaku, 374 TG8120) The hall measurement was done in a commercial system ((Toyo, ResiTest8300)).”
- Line 363: 4mm*4mm*100... * to be replaced with ×.

Response_8: Thanks a lot for pointing out the errors. We have corrected all of them. All the revisions made in the manuscript are marked in red color.

REVIEWERS' COMMENTS:

Reviewer #1 (Remarks to the Author):

The authors have provided a detailed response to the points raised in the review of the first version of this manuscript. The comparison of the proton nmr spectrum of the intercalated phase with that of the salt of the corresponding amine (mis-labelled as NH_4HCl in the response), provides a degree of reassurance that the intercalated amine is present in the protonated form, which then allows a direct comparison between composition and degree of charge transfer. These data together with the additional Supplementary Information that has been added have improved the manuscript considerably and have addressed the technical issues that were raised in the report. The authors' point regarding the doubling of the power factor is well made, although ultimately even though the potential power output is increased, the efficiency will be little changed from that of the earlier hexylammonium intercalate. However, on balance, the improvements that have been made are sufficient to enable publication.

Reviewer #2 (Remarks to the Author):

The authors have completely addressed my comments by illustrations and revisions and therefore I am pleased to recommend the paper for publication in the present form.

Response to Reviewer 1

Comment: *The authors have provided a detailed response to the points raised in the review of the first version of this manuscript. The comparison of the proton nmr spectrum of the intercalated phase with that of the salt of the corresponding amine (mis-labelled as $NH_4.HCl$ in the response), provides a degree of reassurance that the intercalated amine is present in the protonated form, which then allows a direct comparison between composition and degree of charge transfer. These data together with the additional Supplementary Information that has been added have improved the manuscript considerably and have addressed the technical issues that were raised in the report. The authors' point regarding the doubling of the power factor is well made, although ultimately even though the potential power output is increased, the efficiency will be little changed from that of the earlier hexylammonium intercalate. However, on balance, the improvements that have been made are sufficient to enable publication.*

Response: Thank you for your positive comment! The mistake of " $NH_4.HCl$ " has been corrected in caption of supplementary figure 4.

Response to Reviewer 2

Comment: *The authors have completely addressed my comments by illustrations and revisions and therefore I am pleased to recommend the paper for publication in the present form.*

Response: Thank you for your positive comments!